# Acrylic Bone Cements Modified with Graphene Oxide: Mechanical, Physical, and Antibacterial Properties

**DOI:** 10.3390/polym12081773

**Published:** 2020-08-07

**Authors:** Mayra Eliana Valencia Zapata, Lina Marcela Ruiz Rojas, José Herminsul Mina Hernández, Johannes Delgado-Ospina, Carlos David Grande Tovar

**Affiliations:** 1Grupo de Materiales Compuestos, Escuela de Ingeniería de Materiales, Universidad del Valle, Calle 13 #, Cali 100-00, Colombia; valencia.mayra@correounivalle.edu.co (M.E.V.Z.); ruiz.lina@correounivalle.edu.co (L.M.R.R.); jose.mina@correounivalle.edu.co (J.H.M.H.); 2Grupo de Investigación Biotecnología, Facultad de Ingeniería, Universidad de San Buenaventura Cali, Carrera 122 # 6-65, Cali 76001, Colombia; jdelgado1@usbcali.edu.co; 3Faculty of Bioscience and Technology for Food, Agriculture and Environment, University of Teramo, Via R. Balzarini 1, 64100 Teramo, Italy; 4Grupo de Investigación de Fotoquímica y Fotobiología, Universidad del Atlántico, Carrera 30 Número 8-49, Puerto Colombia 081008, Colombia

**Keywords:** acrylic bone cement, antibacterial activity, graphene oxide, mechanical properties, physical properties

## Abstract

Bacterial infections are a common complication after total joint replacements (TJRs), the treatment of which is usually based on the application of antibiotic-loaded cements; however, owing to the increase in antibiotic-resistant microorganisms, the possibility of studying new antibacterial agents in acrylic bone cements (ABCs) is open. In this study, the antibacterial effect of formulations of ABCs loaded with graphene oxide (GO) between 0 and 0.5 wt.% was evaluated against *Gram-positive* bacteria: *Bacillus cereus* and *Staphylococcus aureus*, and *Gram-negative* ones: *Salmonella enterica* and *Escherichia coli*. It was found that the effect of GO was dependent on the concentration and type of bacteria: GO loadings ≥0.2 wt.% presented total inhibition of *Gram-negative* bacteria, while GO loadings ≥0.3 wt.% was necessary to achieve the same effect with *Gram-positives* bacteria. Additionally, the evaluation of some physical and mechanical properties showed that the presence of GO in cement formulations increased wettability by 17%, reduced maximum temperature during polymerization by 19%, increased setting time by 40%, and increased compressive and flexural mechanical properties by up to 17%, all of which are desirable behaviors in ABCs. The formulation of ABC loading with 0.3 wt.% GO showed great potential for use as a bone cement with antibacterial properties.

## 1. Introduction

Acrylic bone cements (ABCs) are employed widely in arthroplasties as a fixation agent between the bone and the implant [1]. Polymethylmethacrylate (PMMA) bone cements consisted of two components: solid and liquid. The solid phase consists mainly of a polymer-based on PMMA, a radio-opaque agent such as barium sulfate, and benzoyl peroxide as the initiator for the polymerization reaction. The liquid phase consists of a methyl methacrylate monomer and a polymerization reaction activator such as *N*,*N*-Dimethyl-*p*-Toluidine. When the liquid and solid phases are mixed, the polymerization reaction of the MMA begins, which causes the cement to cure [2,3,4].

Like all implant materials, bone cements have a high risk of infection when they are implanted in the human body owing to the possibility of biofilms’ formation of microorganisms on an inert surface, which usually requires multiple surgeries for treatment [5].

Chronic infection of articular prosthesis requires surgical removal of the implant in order to eradicate the infectious process. The procedure can be performed in one- and two-stage methods. In the first case, the intervention consists of the removal of the infected implant, decontamination of the site, and implantation of a new revision prosthesis covered with antibiotic-loaded bone cement. In the second case, the placement phase of the revision prosthesis is delayed for several months (2–5 months), during which a bone spacer cement impregnated with an antibiotic is placed in the site [6].

In recent years, increased resistance of microorganisms to antibiotics has led to severe health problems, owing to most of the bacteria causing the infection being resistant to at least one of the antibiotics generally used to eradicate the disease. Therefore, the requirement of significant efforts worldwide to study new antimicrobial agents that can effectively inhibit the growth of bacteria is evident [7]. Furthermore, it is known that, in cements loaded with antibiotics, most of the antibiotic is trapped inside the cement and only a small amount on the surface is available for diffusion to the desired site [5].

Graphene oxide (GO) is essentially a sheet of graphene with a random distribution of oxygen-rich functionalities on its surface. It is characterized by its high performance mechanical [8], biocompatibility [9], biodegradability [10,11], and excellent antimicrobial properties [7,9,12].

Studies reported that the addition of GO to ABCs resulted in increased mechanical properties in compression [13], bending [14], fracture toughness, fatigue [15], and wear [16]; and increased biocompatibility with MC3-T3 cells [17], human bone marrow mesenchymal stem cells (hBMSCs) [18], mouse L929 fibroblasts, and human Saos-2 [19]. There are also studies about modifications to GO by silanization to improve the mechanical properties of cements [20]. Additionally, Tavakoli et al. [2] evaluated the addition of a chitosan/GO composite to increase the mechanical properties, handling, and injectability of the cement, and to improve the cell viability, growth, and cell adhesion to MG-63 cell culture.

Despite the results shown above, the antimicrobial activity of ABCs with GO for use in orthopedics is a little-explored field. Few studies are evaluating this topic. Paz et al. [17] evaluated the antimicrobial activity of ABCs loadings with 0.1 wt.% against *S. aureus* and found no antibacterial effect, possibly owing to the low concentration used. Additionally, our group has evaluated the antibacterial activity against *E. coli*, reporting promising results with 0.3 wt.% GO [16].

In the present study, the effect of GO incorporation between 0 and 0.5 wt.% on the antibacterial activity of ABCs was evaluated against *Gram-positive* (*B. cereus* and *S. aureus*) and *Gram-negative* (*S. enterica* and *E. coli*). Properties such as roughness, handling, residual monomer content, water contact angle, strength, modulus of compression, and bending were studied to evaluate the possible application of these cements with antibacterial properties in orthopedics.

## 2. Materials and Methods

### 2.1. Materials

The solid phase was composed of PMMA beads (New Stetic SA, Medellin, Colombia), barium sulfate (BaSO_4_) (Alfa Aesar, Tewksbury, MA, USA), and benzoyl peroxide (BPO) (Sigma-Aldrich, Palo Alto, CA, USA). The liquid phase was composed of methyl methacrylate (MMA), 2-(diethylamino) ethyl acrylate (DEAEA), 2-(diethylamino) ethyl methacrylate (DEAEM) (Sigma-Aldrich, Palo Alto, CA, USA), *N*,*N*-dimethyl *p*-toluidine (DMPT) (Merck, Burlington, MA, USA), and GO synthetized by a Hummers method modified.

### 2.2. Synthesis and Characterization of GO

GO was synthesized by the Hummers method modified as reported by Valencia et al. [16]. Briefly, 3.0 g of graphite was oxidized with KMnO_4_ for three days and then washed with ethanol and Milli Q water in a centrifuge. The sediment was purified by dialysis tube (Thermo Scientific, Suwanee, GA, US) and then lyophilized at −51 °C and 0.12 mBar (Labconco, Kansas City, MO, US).

GO was characterized by Fourier transform infrared in attenuated total internal reflectance mode (ATR-FTIR) (Perkin-Elmer, Waltham, MA, US); X-ray photoelectron spectra (XPS) (Specs, Berlin, Germany) with a PHOIBOS 150 1D-DLD analyzer (PHOIBOS, Kowloon, Hong Kong, China), using a monochromatic source of Al-Kα (1486.7 eV, 13 kV, 100 W), C1s spectrum from GO was analyzed, and dynamic light scattering (DLS) in a Zetasizer Nano ZS DLS (Malvern Panalytical, JarmanWay, Royston, UK).

### 2.3. Preparation of ABCs

ABCs consisted of two phases (liquid and solid). The solid phase (SP) was composed of 88 wt.% of PMMA, 10.0 wt.% of BaSO_4_, and 2.0 wt.% of BPO. Constituents of the liquid phase (LP) are shown in Table 1. For each formulation, the SP/LP ratio was 2.0. LP after stay 1 h in ultrasound to disperse the GO (Figure 1) was added to the SP and manually mixed. Then, the dough was placed in Teflon molds with the dimensions specified for each test.

### 2.4. Characterization of ABCs

#### 2.4.1. Fourier Transform Infrared (FTIR)

The chemical composition of ABCs specimens was studied by Fourier transform infrared in attenuated total internal reflectance mode (ATR-FTIR) (Perkin-Elmer, Waltham, MA, US).

#### 2.4.2. Thermogravimetric Analysis (TGA)

TGA of ABCs specimens was performed in a TGA Q500 analyzer (TA Instruments, New Castle, DE, US), between 50 and 500 °C, at a heating rate of 10 °C/min.

#### 2.4.3. Scanning Electron Microscope (SEM)

The surface morphology of the specimens fractured during the compression test was coated with gold and analyzed by an environmental scanning electron microscope (ESEM) Philips XL30 (Philips, Eindhoven, The Netherlands).

#### 2.4.4. Water Contact Angle (WCA)

The water contact angle (WCA) was measured at 25 °C using a KSV CAM200 tensiometer (KSV Instruments, Helsinki, Finland). At least ten separate measurements were taken on each sample.

#### 2.4.5. Handling Properties

The maximum temperature reached during polymerization (T_max_) and setting time (t_set_) was determined by the procedure specified in ISO 5833-02 [21]. The cement dough was deposited in Teflon molds of 68 mm diameter and 20 mm height, the temperature and time were recorded, and T_max_ and t_set_ were calculated from the temperature versus time plot. The test was performed in duplicate.

#### 2.4.6. Mechanical Properties

Four-point bending and compression tests were conducted following the specifications of the ISO 5833-02 [21] in a universal testing machine Tinius Olsen-H50KS (Tinius Olsen, Redhill, United Kingdom). Specimens for bending and compression consisted of 75 mm × 10 mm × 3 mm beams and 6 mm diameter and 12 mm high cylinders, respectively. The bending test was conducted at a crosshead displacement rate of 5 mm/min, while the compression test was conducted at 20 mm/min. For each of these tests, a minimum of six specimens was tested for each of the ABCs’ formulations.

#### 2.4.7. Dynamic Mechanical Analysis (DMA)

The dynamic mechanical analysis was carried out in a DMA Q800 (TA Instruments, New Castle, DE, US) in Dual Cantilever mode to 1 Hz of frequency in a range from −100 to 200 °C at a heating rate of 3 °C/min. Specimens consisted of 46 mm × 6 mm × 4 mm beams. *Tg* was defined as the temperature where the maximum of tan δ occurs. At least three specimens were tested per formulation.

#### 2.4.8. Residual Monomer Content (RMC)

Proton nuclear magnetic resonance spectra (^1^H-NMR) were conducted in a Bruker Avance III HD-400 spectrometer (Bruker, Billerica, MA, US) at 25 °C. Specimens were dissolved in deuterated chloroform (CDCl_3_) one week after being prepared. Residual monomer content (*RMC*) in a sample was calculated by integrating the signals of the methoxy protons of the *MMA* (δ = 3.7 ppm) and the *PMMA* (δ = 3.5 ppm) using Equation (1).
(1)$$\mathrm{RMC}\left(\%\right)=\frac{A_{MMA}}{A_{MMA}+A_{PMMA}}\times 100$$
where *A_MMA_* is the signal area of methoxy protons of *MMA* and *A_PMMA_* is the signal area of methoxy protons of *PMMA*.

#### 2.4.9. Antimicrobial Activity

The antimicrobial activity of ABC was determined against *Gram-positive* bacteria *Bacillus cereus* (ATCC 13061) and *Staphylococcus aureus* (ATCC 55804), and *Gram-negative* bacteria *Salmonella enterica* (ATCC 13311) and *Escherichia coli* (ATCC 11775). The inoculums were obtained from cultures in liquid medium of 24 h of growth and adjusted to a concentration of 10^6^ colony-forming units (CFU) mL^−1^ in a spectrophotometer Eppendorf BioSpectrometer^®^ kinetic at 600 nm (Eppendorf, Hamburg, Germany). The dilutions were made utilizing cell washing, centrifuging the strains at 4000 rpm, and resuspending the pellet obtained in nutrient broth diluted 1 in 500.

The sterile ABCs were deposited in microtubes together with 1000 μL of each of the inoculum and placed in agitation at 250 rpm on an IKA KS 3000i orbital shaker (IKA, Staufen, Germany) for 24 h at 37 °C. The control was performed without the ABCs.

Inhibition of cell activity was determined by adding 20 μL of 0.2% of 2,3,5-triphenyltetrazolium chloride (TTC) to an aliquot of 100 μL of the strain exposed to the treatments in a microplate well and incubated for 2 h at 37 °C. Turning the coloring of the well contents to red indicated the metabolic activity of the pathogen. Inhibition of the activity was tested by direct seeding of 100 μL of each of the treatments by making serial dilutions on Müller Hinton agar and incubating at 37 °C for 24 h. The tests were conducted in triplicate.

### 2.5. Statistical Analysis

Quantitative properties are shown as the mean ± standard deviation. For each property, the Student *t*-test compared the means for significant differences. The difference was considered statistically significant when *p* < 0.05. Statistical analysis was performed comparing ABCs loaded of GO with respect to ABCs without GO.

## 3. Results and Discussion

### 3.1. GO Characterization

FTIR characterization of synthesized graphene oxide by the modified Hummers method [16] showed the characteristic bands of this material reported by other authors (Figure 2a). At 3369 and 1046 cm^−1^, the stretching vibrations of the hydroxyl functional group; at 1729 cm^−1^, the vibration of the carbonyl group (C=O); at 1620 cm^−1^, the vibration of the double bond C=C of the aromatic structure of GO; at 1374 cm^−1^, the vibration of the carboxyl group; and at 1224 cm^−1^, the characteristic peak of the epoxy group [20,22,23,24,25]. DLS results (Figure 2b) show an average particle size of 400 nm for GO sheets.

The deconvolution of the C(1s) spectrum of GO in Figure 2c confirms the presence of the functional groups shown in the FTIR spectrum. The results of the binding energies shown in Table 2 allowed quantification of the atomic percentage of these groups and determined the level of oxidation of GO at 67.84% [26]. This degree of oxidation and the O1s/C1s ratio from 1.06 indicates that the graphite oxidation process was successful, producing functional groups such as –COOH, –C=O, and C–OH on the surface of the graphitic structure. Additionally, the presence of the carboxyl group (2.46 at%) could favor the chemical interactions with PMMA and the antimicrobial activity of the cements added with this nanomaterial, because part of the antimicrobial activity of GO is based on the presence of these carboxylic groups at its edges [27,28,29].

### 3.2. Characterization of ABCs

#### 3.2.1. FTIR

Figure 3a shows the infrared spectrum of bone cements. The characteristic peaks of PMMA at 2995, 1723, 1435, and 1142 cm^−1^ corresponding to C–H stretching, C=O ester carbonyl stretching, C–H bending, and C–O stretching vibrations are shown [31,32,33]. The presence of GO in the formulations generated weak absorption peaks at 3752 cm^−1^ in the formulation with 0.5 wt.% of GO (Figure 3b), corresponding to the vibration of the O–H groups present in the GO. The presence of this new peak in the spectrum without changes in the characteristic peaks of PMMA confirmed physical interaction with the GO and the ABC matrix, rather than confirming the secondary hydrogen bridge links between GO and the ABC [34].

#### 3.2.2. TGA

The thermogravimetric analysis of the ABC formulations studied is evident in Figure 4. The graph indicates three stages in the thermal degradation of ABCs that coincide with those reported by other authors for PMMA [33,35]. These stages are related to the degradation of each type of radical structures formed during the polymerization of PMMA [35]. The first one with a 5% mass loss from 145 to 264 °C corresponds to water evaporation and chain fragmentation of the chains formed by the union of two identical radical-chain halves in growth that are known as a head-to-head attachment. The second phase (10% mass loss) occurs between 264 and 320 °C with double bond cleavages of allyl vinylidene in the final position. The third phase between 320 and 440 °C, with a 78% mass loss, corresponds to thermal depolymerization owing to the cleavage of the head–tail bonds [14].

The thermal stability of the cements is reduced with the addition of GO, because, from the second mass loss phase (264 °C), it is observed that the formulation without GO showed a lower weight loss compared with those containing GO. Furthermore, according to the derivate-TGA curve (DTGA), the temperature with the highest mass loss rate decreases from 382 °C for the formulation without GO to 375 °C for the formulation with 0.5% GO. This decrease in the thermal stability of ABCs may be related to the ability of GO to capture free radicals during polymerization, thus generating slightly shorter chains with fewer secondary links, which are partly responsible for the thermal stability [26,36].

#### 3.2.3. SEM

According to Figure 5, which shows the fracture zone of compression specimens of GO cement formulations, the increase in GO content generates an increase in ABC roughness that is more evident in the formulations with 0.3 and 0.5 wt.%, which can be beneficial in facilitating cell anchorage [16,18]. Also, some authors suggest that the surface roughness of biomaterials, such as Ti and PMMA, show a direct relationship with cell adhesion, proliferation, and differentiation [37,38].

Figure 6a shows that GO films are extended in the form of sheets in the PMMA matrix and presented excellent adhesion to the polymer matrix (Figure 6b). The good dispersion and adhesion of GO in the PMMA matrix is beneficial for improving the mechanical properties of ABCs.

#### 3.2.4. Water Contact Angle (WCA)

Higher contact angles indicate lower surface wettability. Figure 7 shows that wettability in ABCs has a direct relationship with GO content, with an increase of 16.6% in the formulation at 0.5 wt.% GO. This characteristic that confers GO to the surface of the ABC is because of its hydrophilic nature, generated by the presence of hydroxyl, epoxy, carbonyl, and carboxyl functional groups [39].

The hydrophilicity of the materials has been related to increased adhesion of osteoblasts to the surface, which promotes osteogenesis [40,41]. Besides, some studies suggest that the presence of more hydrophilic surfaces in PMMA favors antimicrobial activity owing to the formation of a hydration layer that can bind to the surrounding water and generate a physical and energetic barrier that prevents microbial adhesion [42,43].

#### 3.2.5. Handling Properties

According to Figure 8, all the ABCs’ formulations studied reached, during polymerization, T_max_ lower than 90 °C and t_set_ between 3 and 15 min (180 and 900 s), which are values allowed by ISO 5833-02 [21]. In general, T_max_ decreased and t_set_ increased with increasing GO content, mainly owing to the inhibitory and retarding effect of GO on the polymerization reaction as it acts as a sink for free radicals [15,26,44].

It has been widely reported that high polymerization temperatures generate thermal necrosis of the tissue surrounding the cement, which leads to loosening of the prosthesis [3,15,45,46,47], and that higher t_set_ values within the limits of the standard give the surgeon more time to conform the cement during the operation [48]. The t_set_ is an essential clinical requirement in applications such as vertebroplasty and kyphoplasty, where the cement is injected through the vertebra. On the other hand, prolonged t_set_ facilitates the manageability of the ABC during the procedure [49,50,51].

#### 3.2.6. Mechanical Properties

The results of the mechanical characterization of the ABCs are presented in Figure 9. ISO 5833-02 sets the minimum values for compressive strength, flexural strength, and modulus of bending, as 70 MPa, 50 MPa, and 1800 MPa, respectively. These minimum values are highlighted in Figure 9 with a red dashed line. All formulations loaded with GO complied with the mechanical requirements, except the formulation with 0.5% GO, which did not reach the bending strength value.

Compressive strength and compression modulus were significantly improved (*p* < 0.01 and *p* < 0.05, respectively) with GO contents up to 0.2 wt.%, while flexural strength showed a reinforcement up to GO contents of 0.3 wt.%. The GO content does not seem to have any significant effect on the flexural modulus.

Mechanical reinforcement of GO is probably owing to a homogeneous dispersion of the nanoparticles in the matrix [16]. Furthermore, the increased chemical interaction at the PMMA/GO interface owing to the high surface area of the GO sheets (Figure 6a) improves interfacial adhesion by forming hydrogen bridges between the hydroxyl (OH) groups of GO and the methoxyl (OCH_3_) groups of PMMA [2,26,52]. The use of GO loads > 0.3 wt.% resulted in detrimental mechanical properties, possibly owing to the formation of GO agglomerates that act as stress concentrators [15].

#### 3.2.7. DMA and RMC

Figure 10 shows the results of the storage module (E’) and tan δ of the ABCs loaded with GO as a function of temperature. E’ increased with intermediate GO percentages (0.2 and 0.3%) and decreased with the lowest and highest GO percentages (0.1 and 0.5 wt.%). The previous result indicates that, possibly, the best dispersion of GO is achieved with the average rates of GO, which is corroborated with the mechanical properties to bending.

The *Tg* of the ABCs, obtained from the tan δ graph (Table 3), shows a slight increase for cements with GO (approximately 4%) compared with the sample without GO. This is possibly owing to the mechanical reinforcement generated by the nanoparticles and the chemical interactions with the matrix [16].

The residual monomer content shows a slight increase in the two formulations with higher GO content (Table 3). These RMC results, together with the decrease in T_max_ and the rise in t_set_ during curing, confirm the retarding and inhibiting effect that GO generates during the free radical polymerization reaction [15].

#### 3.2.8. Antibacterial Activity

According to the results shown in Figure 11, ABCs without GO showed weak antimicrobial activity against *gram-positive* (25%) and *gram-negative* (21%) bacteria, such inhibition may be related to the release or interaction at the surface with unpolymerized MMA (RMC 1.06%), which may generate an inhibition by the susceptibility of the bacteria to traces of residual monomer. Moreover, the percentages of GO > 0.1 wt.% in the ABC formulation have a significant effect (*p* < 0.01) in decreasing the number of colony-forming units in both *Gram-positive* and *Gram-negative* bacteria. It was found that, by increasing the concentration of GO in the study range, the concentration of CFU for *Gram-positive* (r^2^ > 0.96) and *Gram-negative* (r^2^ > 0.82) bacteria was proportionally decreased linearly, achieving a reduction of 63 and 76% for *Gram-positive* and *Gram-negative* CFU, respectively.

These results differ from those reported in CS/PVA/GO films [53], where *Gram-negative* bacteria had higher resistance to inhibition, suggesting inhibition against *Gram-negative* or *Gram-positive* bacteria, showing GO is dependent on the matrix in which it is included. It is found that the inhibitory effect against *S. aureus* bacteria is slightly higher compared with *B. cereus* (Figure 11a) and that the impact on *E. coli* bacteria is also more significant when compared with *S. enterica* (Figure 11b).

In Table 4, it is observed that the inhibition of the cellular activity of GO is higher on *Gram-negative* bacteria because a total inhibition of the pathogen is achieved with a lower percentage of GO in the cement (GO = 0.2 wt.%) compared with the *Gram-positive* bacteria studied, where ABCs’ loadings with 0.3 wt.% GO are necessary to achieve total inhibition. This could be attributed to different cell wall structures, which further cause different GO penetration probability. Once inside, the sharp ends will puncture the membrane and produce irreversible damage to the cells [54]. Another possibility is the generation of reactive oxygen species, which causes oxidative stress and cell membrane destruction after direct contact [55].

The low antibacterial activity of cements with 0.1 wt.% GO is consistent with the results reported by Paz et al. They found no antibacterial activity against *S. aureus* in ABCs loaded with the same amount of GO. That was probably because of the low concentration used [17], as the antibacterial activity of GO has been widely reported as dose-dependent [27,29].

Several mechanisms have been suggested for bacterial inactivation. The first involves direct contact of the bacteria with the extremely sharp edges of the GO, which causes physical damage to the membrane. The second mechanism consists of the generation of oxidative stress on the membrane by the production of reactive oxygen species (ROS), which leads to cell death [9,27,56,57]. Further studies are required to elucidate the possible antibacterial mechanisms presented in ABCs loading with GO to predict how it will behave against other types of bacteria.

Owing to the antibacterial mechanisms reported for GO, it is evident that the high degree of oxidation reached by the graphite during synthesis (71.2%) and the correct exfoliation that led to the obtaining of GO of a few nanometers of thickness (9 nm) favored the antibacterial activity of the studied ABCs.

## 4. Conclusions

GO was obtained with an oxidation degree of 71.2% and an average lateral size of 400 nm. These characteristics of the nanomaterial allowed to get excellent dispersibility in the liquid phase and an increase in mechanical, physical, and antibacterial properties.

The presence of GO presented the following desirable properties in the ABCs: an increase in roughness, in wettability (by 17%), in t_set_ (by 40%), in the mechanical properties to compression and flexion up to 17%, and in the antibacterial activity against *Gram-positive* and *Gram-negative* bacteria studied, in addition to a decrease in T_max_ up to 19%. However, it also generated a slight reduction in thermal stability and a negligible increase in residual monomer content.

Because the formulation of ABC with 0.3 wt.% GO presented the minimum concentration of GO with which it is possible to inactivate both *Gram-positive* and *Gram-negative* bacteria studied, in addition to complying with the mechanical and handling requirements established in ISO 5833-02, this formulation is suggested to develop complementary studies that confirm its high potential for use as a replacement for bone cements loaded with antibiotics as a solution in the treatment of antibiotic-resistant bacteria in orthopedics.

## Figures and Tables

**Figure 1 polymers-12-01773-f001:**
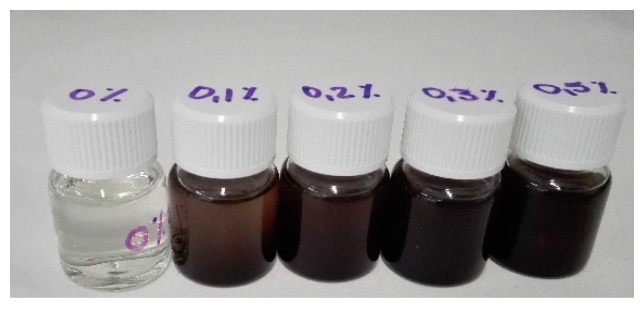
Photographs of the liquid phase of the acrylic bone cements’ (ABCs) formulations. From left to right, the percentage of graphene oxide (GO) increases from 0 to 0.5 wt.%.

**Figure 2 polymers-12-01773-f002:**
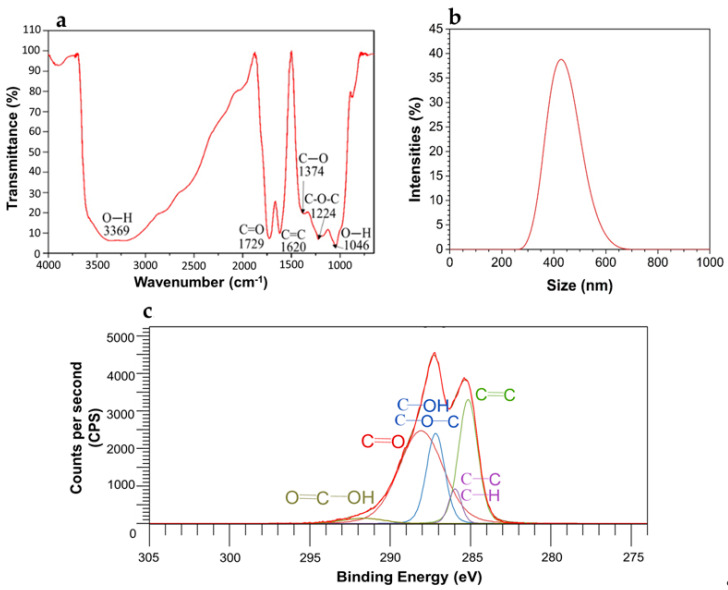
Results characterization of GO. (**a**) Fourier transform infrared (FTIR) spectrum, (**b**) Dynamic light scattering (DLS), and (**c**) Peak deconvolution of C(1s) in the X-ray photoelectron spectra (XPS) of the graphene oxide.

**Figure 3 polymers-12-01773-f003:**
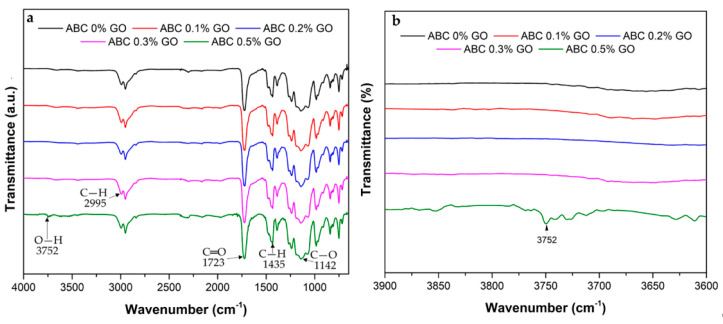
FTIR spectra of ABCs loading with different percentages of GO. (**a**) Spectrum from 700 to 4000 cm^−1^ and (**b**) spectrum from 3600 to 3900 cm^−1^.

**Figure 4 polymers-12-01773-f004:**
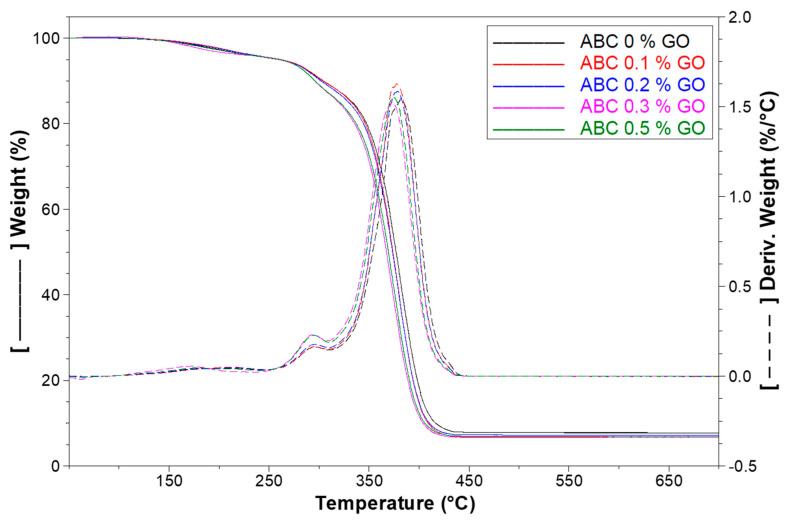
Thermogravimetric analysis (TGA) and Derivate-TGA(DTGA) thermograms of ABCs loading with different percentages of GO.

**Figure 5 polymers-12-01773-f005:**
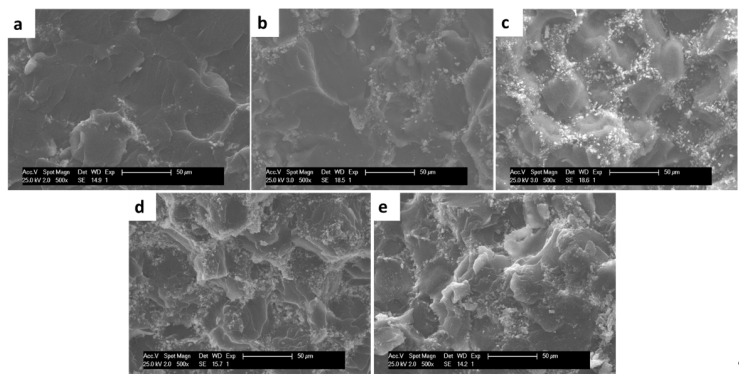
SEM images of the fractured surface of samples to the compression test of ABC modified with GO. (500×). (**a**) ABC 0% GO, (**b**) ABC 0.1% GO, (**c**) ABC 0.2% GO, (**d**) ABC 0.3% GO, and (**e**) ABC 0.5% GO.

**Figure 6 polymers-12-01773-f006:**
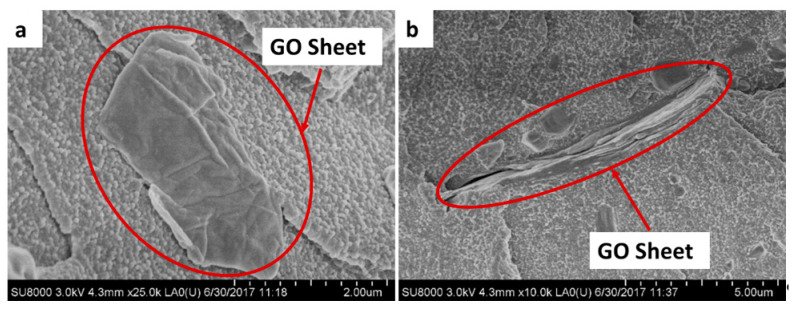
Scanning electron microscope (SEM) images of ABC with 0.5 wt.% GO, where the GO sheets are shown in the cement at different magnifications. (**a**) 25,000× and (**b**) 10,000×.

**Figure 7 polymers-12-01773-f007:**
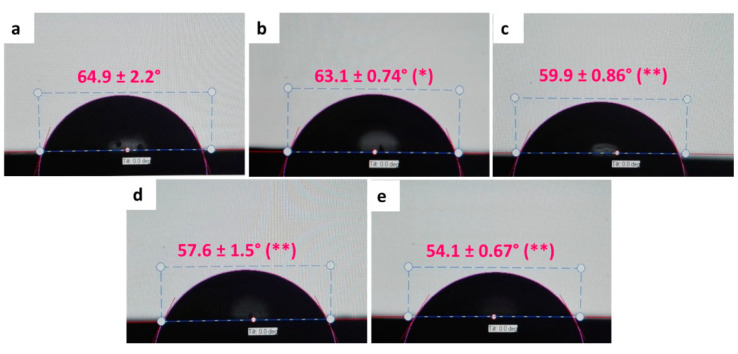
Photographs of water contact angle (WCA) on the surfaces of ABC with different percentages of GO. (**a**) ABC 0% GO, (**b**) ABC 0.1% GO, (**c**) ABC 0.2% GO, (**d**) ABC 0.3% GO, and (**e**) ABC 0.5% GO. Asterisks denote significant differences of ABC loading samples respect to ABC 0% GO at a significance levels of * *p* < 0.05 and ** *p* < 0.01.

**Figure 8 polymers-12-01773-f008:**
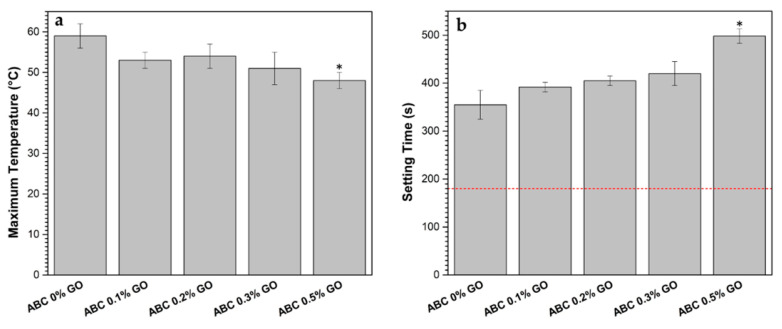
Handling parameters of liquid–powder mixture in ABC loadings with GO. (**a**) The maximum temperature reaches during the polymerization and (**b**) setting time of each formulation.The red dash line indicates the minimum value of the property set in ISO 5833-02 [21]. Asterisks denote significant differences of ABC loading samples with respect to ABC 0% GO at a significant level of * *p* < 0.05.

**Figure 9 polymers-12-01773-f009:**
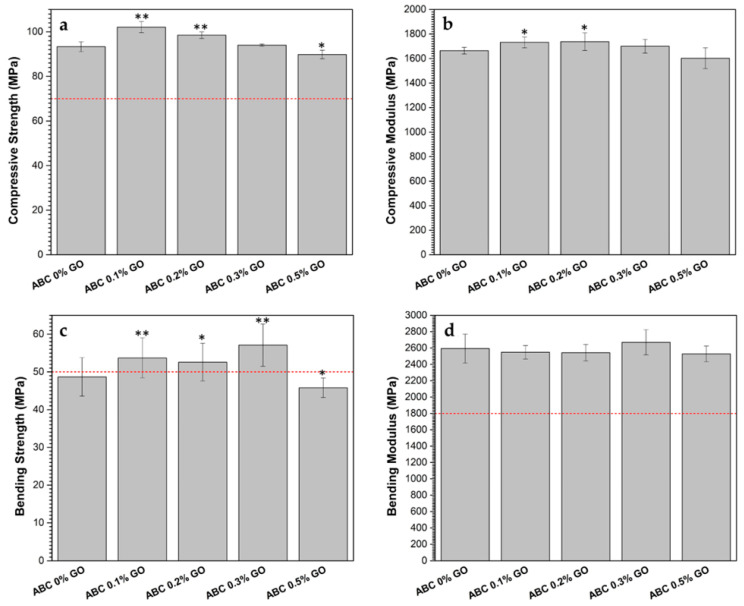
Mechanical properties compression and bending in ABC modified with different percentages of GO. (**a**) Compressive strength, (**b**) compressive modulus, (**c**) bending strength, and (**d**) bending modulus. The red dash line indicates the minimum value of the property set in ISO 5833-02 [21]. Asterisks denote significant differences of modified ABC samples respect to ABC 0% GO at significance levels of * *p* < 0.05 and ** *p* < 0.01.

**Figure 10 polymers-12-01773-f010:**
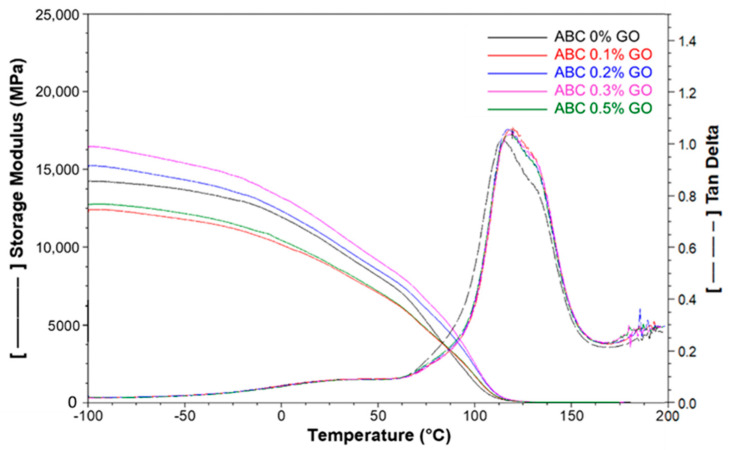
Storage modulus and tangent delta de ABC modified with GO obtained by dynamic mechanical analysis (DMA).

**Figure 11 polymers-12-01773-f011:**
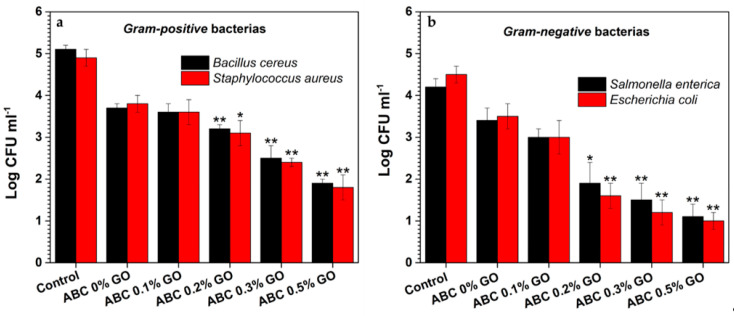
Colony-forming units (CFU) of *Gram-positive* (**a**) and *Gram-negative* strains (**b**) after 24 h incubation with GO-modified ABCs. Asterisks denote significant differences of ABC loading samples with respect to ABC 0% GO at significance levels of * *p* < 0.05 and ** *p* < 0.01.

**Table 1 polymers-12-01773-t001:** Composition (wt.%) of the liquid phase of the acrylic bone cements’ (ABCs) formulations. GO, graphene oxide; MMA, methyl methacrylate; DEAEA, 2-(diethylamino) ethyl acrylate; DEAEM, 2-(diethylamino) ethyl methacrylate; DMPT, *N*,*N*-dimethyl *p*-toluidine.

Formulation	MMA	DEAEA	DEAEM	DMPT	GO
ABC 0% GO	88	1	1	2.5	0
ABC 0.1% GO	83	1	1	2.5	0.1
ABC 0.2% GO	78	1	1	2.5	0.2
ABC 0.3% GO	73	1	1	2.5	0.3
ABC 0.5% GO	68	1	1	2.5	0.5

**Table 2 polymers-12-01773-t002:** Analysis peak deconvolution of C(1s) of the graphene oxide [30].

	Binding Energy (eV)	Atomic Concentration (at.%)
C=C	285.2	27.21
C–C and C–H	286.0	4.95
C–O–C and C–OH	287.2	18.77
C=O	288.1	46.61
O=C–OH	291.7	2.46

**Table 3 polymers-12-01773-t003:** Glass transition temperature (*Tg*) taken as the maximum tangent of delta in dynamic mechanical analysis (DMA) graphic and residual monomer content (RMC) calculated by Proton nuclear magnetic resonance (^1^H-NMR) of ABC modified with different percentages of GO.

Formulation	*Tg* (°C)	RMC (%)
ABC 0% GO	115	1.06
ABC 0.1% GO	118	1.06
ABC 0.2% GO	120	1.06
ABC 0.3% GO	120	1.32
ABC 0.5% GO	119	1.32

**Table 4 polymers-12-01773-t004:** Inhibition of the cellular activity of cement added with GO against four bacterial strains.

Strain	Control	ABC 0% GO	ABC 0.1% GO	ABC 0.2% GO	ABC 0.3% GO	ABC 0.5% GO
*Bacillus cereus* ATCC 13061	+++	+	+	+	-	-
*Staphylococcus aureus* ATCC 55804	+++	+	+	+	-	-
*Salmonella entérica* ATCC 13311	+++	+	+	-	-	-
*Escherichia coli* ATCC 11775	+++	+	+	-	-	-

(+++) Strong activity of the pathogen, represented by a turning red of the TTC; (+) Weak activity of the pathogen, represented by a turning pink of the TTC; (-) Without the activity of the pathogen, represented by the absence of turn of the TTC; TTC: 2,3,5-Triphenyltetrazolium chloride, an indicator of dehydrogenase enzyme activity.

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
