# Peer review of "Acrylic Bone Cements Modified with Graphene Oxide: Mechanical, Physical, and Antibacterial Properties"

_polymers, 2020, doi:10.3390/polym12081773_

Round 1
Reviewer 1 Report
Article ”Acrylic Bone Cements Modified with Graphene Oxide: Mechanical, Physical, and Antibacterial Properties” is written well and basically could be recommended for publication. Bellow I propose some small text corrections which can be fixed by adding some explanations or removing selected sentences.
Fig 2 XPS result “y“ axis label I assume cps is counts per second please expand this shortcut in the Fig or in the label bellow.
After sentence “These stages are related to the degradation of each type of radical structures formed during the polymerization of PMMA” (212 line) please explain radial structures or add citation. The same in line 223
Sentence “This could be attributed to the electron affinity between opposite charges present on the bacterial cell wall and GO sheets [52]” (line 338)
If that is a suggestion that opposite charges attract, then GO has negative charge on the surface what is the surface charge of the bacteria. Maybe the small differences lie simply in different cell wall structure which further cause different GO penetration probability and charge has nothing to do here? Please justify this mechanism by comparison of surface charges.
Please add a comment below the table how did you ascribed strong or weak activity in table 4
Author Response
Fig 2 XPS result "y "axis label I assume cps is counts per second, please expand this shortcut in the Fig or the label bellow.
R// We appreciate the reviewer's comment, and Figure 2 was modified.
After the sentence "These stages are related to the degradation of each type of radical structures formed during the polymerization of PMMA" (212 lines), please explain radial structures or add citations. The same in line 223
R// We appreciate the reviewer's comment, and the references were added after both sentences.
The sentence "This could be attributed to the electron affinity between opposite charges present on the bacterial cell wall and GO sheets [52]" (line 338)
If that is a suggestion that opposite charges attract, then GO has a negative charge on the surface, what is the surface charge of the bacteria. Maybe the small differences lie simply in different cell wall structures, which further cause different GO penetration probability, and the charge has nothing to do here? Please justify this mechanism by comparison of surface charges.
R// We appreciate the reviewer's comment. The sentence "This could be attributed to the electron affinity between opposite charges present on the bacterial cell wall and GO sheets [52]" was modified as "This could be attributed to different cell wall structures, which further cause different GO penetration probability. Once inside, the sharp ends will puncture the membrane and produce irreversible damage to the cells [54]. Another possibility is the generation of reactive oxygen species that causes oxidative stress and cell membrane destruction after direct contact [55]. "
Please add a comment below the table how did you ascribe strong or weak activity in table 4
R// We appreciate the reviewer's comment and a comment was added describing how each level of activity was obtained.
"(+++) Strong activity of the pathogen, represented by a turning red of the TTC.
(+) Weak activity of the pathogen, represented by a turning pink of the TTC.
(-) Without the activity of the pathogen, represented by the absence of turn of the TTC.
TTC: 2,3,5-Triphenyltetrazolium chloride, an indicator of dehydrogenase enzyme activity."

Reviewer 2 Report
This work systematically studied the mechanical, physical and antibacterial properties of GO -ABCs composites. This paper can be accepted after minor revision.
- In line 20-21, the possibility of studying new antibacterial agents in ABCs is open. It should change to : the possibility of studying new antibacterial agents in acrylic bone cements (ABCs) is open.
- In line 111-114, before ESEM observation, how to prepare the specimens? Whether the specimens were coated with carbon particles or gold particle.
- In the section of FTIR, line 199-203, the presence of these new peaks in the spectrum without changes in the characteristic peaks of PMMA confirmed physical interaction the GO and the ABC matrix, rather than confirmed the secondary hydrogen bridge links between GO and the ABC.
Author Response
- In line 20-21, the possibility of studying new antibacterial agents in ABCs is open. It should change to: the possibility of studying new antibacterial agents in acrylic bone cements (ABCs) is open.
R// We appreciate the reviewer's comment, and the sentence was changed.
- In line 111-114, before ESEM observation, how to prepare the specimens? Whether the specimens were coated with carbon particles or gold particle.
R// We appreciate the reviewer's comment, and in the text was added, then the samples were coated with gold.
- In the section of FTIR, lines 199-203, the presence of these new peaks in the spectrum without changes in the characteristic peaks of PMMA confirmed physical interaction the GO and the ABC matrix, rather than confirmed the secondary hydrogen bridge links between GO and the ABC.
R// We appreciate the reviewer's comment, and the text was modified according to the reviewer's suggestion.

Reviewer 3 Report
Valencia Zapata et al. report on the mechanical, physical, and antibacterial properties of a bone cement modified with GO. They characterized the samples with a large set of experimental techniques. On my opinion, this work is appropriate for the journal. Nevertheless, the authors should do some corrections before its publication.
Minor corrections:
· The introduction could be improved.
· In section 2.2. Synthesis and characterization of GO, it is important to mention the template they use for AFM characterization.
· The O1s/C1s ratio obtained by XPS will be useful for the readers.
Mayor corrections:
· Page 5: I agree with the experimental data of C1s. Nevertheless, I do not agree with its fitting. C1s should be fitted as in reference [Carbon 93, 967 (2015) and others]. Accordingly, table 2 (page 6) and the respective text have to be modified.
· Page 5: In order of correctly characterizing GO by AFM, a small quantity of the material has to be deposited on an appropriate substrate. Mica is a good option. There are too much material in figure 2d. On my opinion, the authors should repeat this experiment but with less material. An example can be obtained in ref 24 of their text and in [Chemical Communications 46, 1112 (2010)]
· Page 6: Authors claim: “The presence of GO in the formulations generated weak absorption peaks at 3752 and 3443 cm-1” Nevertheless, I cannot see these peaks in the experimental data showed in figure 3. Moreover, It seems that ABC 0% GO also has some structure in this range. On the other hand, the blue spectra (0.2% GO) seems to be flat in the respective range. The authors should do a new figure 3b with a zoom in of the region between 4000 and 3000 cm-1 to show the respective information.
Author Response
Minor corrections:
- The introduction could be improved.
R// We appreciate the reviewer's comment, and the introduction was modified. One paragraph was added "Acrylic bone cements (ABCs) are employed widely in arthroplasties as a fixation agent between the bone and the implant [1]. Polymethylmethacrylate (PMMA) bone cements consisted of two components: solid and liquid. Solid-phase consists mainly of a polymer-based on PMMA, a radio-opaque agent such as barium sulfate, and a polymerization using benzoyl peroxide as the initiator for the polymerization reaction. The liquid phase consists of a methyl methacrylate monomer and a polymerization reaction activator such as N, N-Dimethyl-p-Toluidine. When the liquid and solid phases are mixed, the polymerization reaction of the MMA begins, which causes the cement to cure [2–4]. "
- In section 2.2. Synthesis and characterization of GO, it is important to mention the template they use for AFM characterization.
R// We appreciate the reviewer's comment and the results of AFM were removed from the paper because the current world situation it is impossible for us to repeat the test that was done abroad.
- The O1s/C1s ratio obtained by XPS will be useful for the readers.
R// We appreciate the reviewer's comment and O1s/C1s ratio obtained by XPS was reported in line 183
Major corrections:
- Page 5: I agree with the experimental data of C1s. Nevertheless, I do not agree with its fitting. C1s should be fitted as in reference [Carbon 93, 967 (2015) and others]. Accordingly, table 2 (page 6) and the respective text have to be modified.
R// We appreciate the reviewer's comment, and the fitting of the C1s spectrum was modified according to reference [Carbon 93, 967 (2015), table 2 and the corresponding text were modified.
Page 5: In order of correctly characterizing GO by AFM, a small quantity of the material has to be deposited on an appropriate substrate. Mica is a good option. There is too much material in figure 2d. On my opinion, the authors should repeat this experiment but with less material. An example can be obtained in ref 24 of their text and in [Chemical Communications 46, 1112 (2010)]
R// We appreciate the reviewer's comment, and the AFM assay was conducted abroad because our research group did not have the equipment to perform the test, and due to the current contingency, we can't send samples to repeat the assay. Hence, the authors decided to remove this figure from the document.
Page 6: Authors claim: "The presence of GO in the formulations generated weak absorption peaks at 3752 and 3443 cm-1" Nevertheless, I cannot see these peaks in the experimental data showed in figure 3. Moreover, It seems that ABC 0% GO also has some structure in this range. On the other hand, the blue spectra (0.2% GO) seems to be flat in the respective range. The authors should do a new figure 3b with a zoom in of the region between 4000 and 3000 cm-1 to show the respective information.
R// We appreciate the reviewer's comment, and Figure 3 was modified. The text about it was changed too: "The presence of GO in the formulations generated weak absorption peaks at 3752 cm-1 in the formulation with 0.5 wt.% of GO (Figure 3b), corresponding to the vibration of the O—H groups present in the GO."

Round 2
Reviewer 3 Report
The authors improved the paper and fixed all my comments. It would be nice to have the AFM characterization but I understand the situation. This work is well suited for its publication in polymers. My suggestion is to accept this paper in its current form.